# Royal Jelly—A Traditional and Natural Remedy for Postmenopausal Symptoms and Aging-Related Pathologies

**DOI:** 10.3390/molecules25143291

**Published:** 2020-07-20

**Authors:** Andreea Bălan, Marius Alexandru Moga, Lorena Dima, Sebastian Toma, Andrea Elena Neculau, Costin Vlad Anastasiu

**Affiliations:** 1Department of Medical and Surgical Specialties, Faculty of Medicine, Transylvania University of Brasov, 500019 Brasov, Romania; dr.andreeabalan@gmail.com (A.B.); moga.og@gmail.com (M.A.M.); canastasiu@gmail.com (C.V.A.); 2Department of Fundamental, Prophylactic and Clinical Sciences, Faculty of Medicine, Transilvania University of Brasov, 500019 Brasov, Romania; lorenadima@yahoo.com

**Keywords:** royal jelly, bioactive properties, menopause symptoms, estrogenic effect, apitherapy, fatty acids

## Abstract

Women’s life stages are based on their reproductive cycle. This cycle begins with menstruation and ends with menopause. Aging is a natural phenomenon that affects all humans, and it is associated with a decrease in the overall function of the organism. In women, aging is related with and starts with menopause. Also, during menopause and postmenopausal period, the risk of various age-related diseases and complaints is higher. For this reason, researchers were pushed to find effective remedies that could promote healthy aging and extended lifespan. Apitherapy is a type of alternative medicine that uses natural products from honeybees, such as honey, propolis, royal jelly, etc. Royal jelly is a natural yellowish-white substance, secreted by both hypopharyngeal and mandibular glands of nurse bees, usually used to feed the queen bees and young worker larvae. Over the centuries, this natural product was considered a gold mine for traditional and natural medicine, due to its miraculous effects. Royal jelly has been used for a long time in commercial medical products. It has been demonstrated to possess a wide range of functional properties, such as: antibacterial, anti-inflammatory, vasodilatative, hypotensive, anticancer, estrogen-like, antihypercholesterolemic, and antioxidant activities. This product is usually used to supplement various diseases such as cardiovascular disease, Alzheimer’s disease, sexual dysfunctions, diabetes or cancer. The main objective of this study is to highlight the effectiveness of royal jelly supplementation in relieving menopause symptoms and aging-related diseases. We also aimed to review the most recent research advances regarding the composition of royal jelly for a better understanding of the effects on human health promotion.

## 1. Introduction

Menopause represents a transitional phase into a woman’s life, during which a wide range of changes in metabolism, physiology, mental, and physical well-being occur [1]. Menopausal symptoms usually have a significant negative impact on the overall quality of life and become a psychosocial impairment. The majority of these symptoms are a consequence of estrogen withdrawal or a prolonged estrogen deficiency [2]. Post-menopausal women are at increased risk of atherosclerosis, cardiovascular events, diabetes, mood disorders such as anxiety and depression, vasomotor symptoms (hot flushes, night sweats) osteoporosis and vaginal atrophy [3]. Conventionally, these problems are treated with hormonal replacement therapy but the risk of undesirable adverse effects such as breast cancer, abnormal uterine bleeding or breast tenderness sometimes outweighs the benefits, which is why more natural alternatives tend to be used [4,5]. Natural therapies for menopausal-related symptoms and complaints include phytoestrogens, which are natural molecules similar to human estrogens, considered to prevent postmenopausal symptoms. Soybeans and red clover are the most well-known natural alternatives to conventional hormone replacement therapy, but their efficiency is still highly debated [6].

Apitherapy is a type of alternative medicine that uses natural products from honeybees, such as honey, propolis, royal jelly, bee bread, and bee venom, whose healing properties are described by centuries [7]. These products also demonstrated their usefulness for the treatment of postmenopausal complaints. Royal jelly, known as “superfood”, is a creamy substance secreted by the mandibular and hypopharyngeal glands of nurse bees, used to feed the queen bees throughout their lifetime and worker bees during the stage of larvae [8]. The consumption of this magical liquid by the queen bees generate many advantages such as their size, which is double in comparison to worker bees, a longer lifespan and a better function of their reproductive system [9]. In the last years, the production of royal jelly in China considerably increased because beekeepers developed a genetic selection of high-producing royal jelly Italian bees [10]. China became the largest producer of royal jelly, and nowadays it possesses almost 90% of all world production. Annually, more than 4000 tons of royal jelly are produced and exported from China to the United States or Europe [11,12].

Although there are many medical records of the usage of royal jelly all over the world, it has been mainly used in Asia. Recent reports indicated a high potential of this natural product to improve human health. Royal jelly’s pharmaceutical properties, from animal models to humans, have been widely investigated. Despite the mechanisms of action are still under research, it attracted the attention of many investigators around the world. Royal jelly improves the reproductive health, neurologic diseases, and possesses several biological properties, such as: antibacterial, vasodilative, anti-inflammatory, hypotensive, anticancer, estrogen-like, antihypercholesterolemic, and antioxidant effects [13]. In the last years, royal jelly was reported as a valuable medicinal agent [7,14] for healthy aging and longevity [15].

In this article, we focus on the studies regarding the biological effects of royal jelly during menopause, and on its mechanisms of action. We aim to bring together all these studies in order to have a better understanding of whether and how royal jelly could be a natural remedy for menopausal complaints and aging-related diseases. Figure 1 illustrates the main biological effects of royal jelly.

## 2. Chemical Composition of Royal Jelly

Royal jelly is a rich source of proteins, peptides, sugars, fatty acids, and other bioactive substances. Its variation generally depends on the biodiversity of the flora, characteristic of each geographic area [16], and the seasonal conditions of feeding. The water content of royal jelly represents 60–70% from the final composition. Royal jelly is an acid liquid and its pH ranges between 3.6 and 4.2 [17].

### 2.1. Proteins and Peptides

Proteins represent, after water, the most dominant components of royal jelly. Almost 50% for the final composition are proteins and various peptides. From these, 80–90% are MRJPs (major royal jelly proteins), also known as apalbumins [18]. MRJPs represent a family of nine major proteins, with molecular weights ranging between 49 and 87 kDa. They are termed MRJP1-MRJP9 and are encoded by nine different genes [19].

MRJP1 is the most dominant among all the major proteins family. It contains a monomeric glycoprotein known as royalactin, which activates p70 S6 kinase. This kinase is responsible for the increase of the juvenile hormone levels, essential for the normal development and function of the ovaries [20].

Most of the health and metabolic benefits of the royal jelly are due to the glycosylated proteins from its composition [21,22]. MRJPs family is not the only proteic compound of royal jelly. It contains in small amounts peptides such as aspimin, royalisine and jelleines (jelleines-I to jelleines-IV) [23]. From these, royalisine and jelleines exert antimicrobial effects and enhance the efficiency of the immune response to common infections.

### 2.2. Amino Acids

Royal jelly contains a wide amount of amino acids, especially essential amino acids [24]. The most expressed free amino acid from royal jelly is lysine (62.43 mg/100 g). This is followed by various quantities of cysteine, proline, and aspartic acid. Recent research [25] showed that long-term dietary supplementation of leucine, isoleucine and valine extracted from royal jelly, enhances the expression of sirtuin 1, mitochondrial biogenesis and reduces the production of reactive oxygen species (ROS) in the skeletal muscles and myocardium. In these conditions, royal jelly administration as a dietary supplement can alleviate age-related muscle dysfunctions.

### 2.3. Sugars

Carbohydrates represent almost 15% of the total composition of royal jelly [26]. The main sugars are glucose and fructose, and together represent almost 90% of the final composition. Depending on the beekeeping foods, botanical origin, season, and bee species, glucose may reach almost 50–70% of the total sugars, while sucrose only accounts for 0.8–3.6%. Other small amounts of oligosaccharides such as erlose, trehalose, ribose, raffinose, gentiobiose or melibose may be found in royal jelly composition [27].

### 2.4. Lipids and Fatty Acids

Lipids represent between 7% and 18% of the content of royal jelly, and 80–85% of these are dicarboxylic acids and hydroxy fatty acids [28]. The most common fatty acids from royal jelly are 10-hydroxy-2-decenoic acid (10H2DA), sebacic acid (SA) and 10-hydroxydecanoic acid (10-HDA). 10-HDA represents 3,5% of freeze-dried royal jelly composition and it is the most stable compound [29]. Moreover, it acts like a potent antibacterial agent that protects bee larvae against bacterial infections from the bee hives [30].

10-HDA was reported as a potent inhibitory agent of matrix metalloproteinases (MMPs), which contribute to tisular aging and cause several inflammatory diseases such as arthritis. SA, 10-HDA, and 10H2DA are able to mediate estrogen signaling by increasing the activity of the estrogen receptors (ER). Suzuki et al. [31] reported that several lipidic compounds isolated from royal jelly (trans-2-decenoic acid, 10-hydroxy-trans-2-decenoic acid, 10-HDA, and 24-methylenecholesterol) inhibited the binding of 17β-estradiol to ERβ. However, they exerted no effect on the binding to ERα. These findings provide strong evidence concerning the estrogenic effect of royal jelly compounds. 

### 2.5. Other Constituents: Vitamins, Minerals, Acetylcholine, Polyphenols

Vitamins and minerals represent 0.8–3% of royal jelly fresh matter. The most abundant vitamin from the composition of royal jelly is vitamin B_5_, followed by niacin and small amounts of vitamin A, C, E, and vitamin B_1_, B_2_, B_6_, B_8_, B_9_, and B_12_ [17]. The major minerals are K, Ca, Na, Mg, Zn, Fe, Cu and Mn, in this descending order. Royal jelly also contains trace elements such as Co, Hb, Ba, W, Cr, Ni, V, Pb, Mo, whose concentrations are highly constant [32].

Royal jelly also contains nucleotides (e.g., guanosine, adenosine, and uridine) and phosphates, such as adenosine monophosphate, adenosine diphosphate, and adenosine triphosphate. Adenosine N1-oxide is an oxidized molecule of the adenosine [33,34], which demonstrated its neurogenic effects and its tropism for the central nervous system. It stimulates neurite outgrowth by inducing PC12 cells differentiation into neural cells [34] and contribute to a normal neuronal development [33]. Acetylcholine, which is known as a neurotransmitter, can also be found in royal jelly composition (1 mg/g dry weight). Its consumption may prevent the development of neurogenerative dysfunctions, such as Alzheimer’s disease [35].

The flavonoids described in royal jelly can be divided into the following five categories: isoflavonoids (genistein and formononetin), flavones (apigenin, chrysin and luteolin), flavonols (kaempherol), flavonones (hesperetin, naringerin, isosakuranetin) and isoflavonoids (genistein, formononetin). The phenolic content of royal jelly is based on organic acids such as octanoic or dodecanoic acid and pinobanksin [36]. Figure 2 illustrates a schematical representation of the royal jelly composition and the main functional activities of its compounds.

## 3. Studies of Royal Jelly as a Beneficial Therapeutic Agent for Postmenopausal Symptoms and Aging-Related Diseases

Hormone replacement therapy is the most common treatment used by the majority of women in order to decrease postmenopausal symptomatology and to improve their quality of life during this period. The most significant disadvantage of this type of treatment is the multitude of side effects, which is why replacing it with natural alternatives has been emphasized. Using nonhormonal methods in order to decrease menopause-related symptoms and to treat various pathologies is known as complementary medicine. It consists of the administration of herbal products, phytoestrogens or food supplements [44].

Royal jelly is a traditional and natural product that, due to its similarity with estrogens [45] is used by postmenopausal women for the improvement and treatment of menopause-related complications and aging-related pathologies. The effects of royal jelly during menopause represent an actual subject and many studies were conducted on this topic. Unfortunately, there is only a small amount of human clinical trials and the main majority of the investigations were realized on ovariectomized animal models. 

### 3.1. Estrogen-like Activity

Royal jelly has shown its estrogenic effect both in vitro and in vivo. This effect is mediated through the interaction with estrogenic receptors (ER). Daily administration of royal jelly endorsed ovarian hormones and follicular development in a rat model and improved the fertility parameters. Royal jelly also improved the oocyte maturation, ameliorated redox status and activated glucose pathways in cumulus oophorus [46]. Besides, a recent study [47] showed that royal jelly administration, in conjunction with exogenous progesterone, similarly increased the pregnancy rate and the estrogenic response in Awassi ewes.

Luteinizing hormone (LH) and folliculo-stimulant hormone (FSH) are gonadal hormones involved in reproduction. Their levels are under estrogen and inhibin control. In addition, these hormones are involved in the natural aging process. In young women, the high amounts of estrogen maintain FSH and LH decreased. In menopause, the reduction of the follicular size, associated with luteal regression, affects the regular release of estrogen and inhibin, leading to abnormal secretion of FSH. The supplemental dietary consumption of royal jelly was reported very useful in this process, due to its content of fatty acids, especially 10-hydroxyl-2-decenoic acid. This compound increases the synthesis of estrogens and maintains low levels of serum FSH and LH. Furthermore, 10-hydroxyl-2-decenoic acid is very efficient in preventing aging-related follicular depletion and enhancing hormonal regulation [48].

The research on ovariectomized rats showed that royal jelly competes with E2 for binding ERα and ERβ, but its affinity is weaker in comparison to phytoestrogens or diethylstilbestrol. Also, royal jelly may restore VEGF expression in the uterus by increasing gene transcription in MCF-7 cells, while 20 mg/kg of 17β-estradiol may restore the expression of this factor in both uterus and brain [45].

Clinical studies showed that the oral administration of royal jelly (1 g/day) is able to decrease the severity of the premenstrual syndrome and to improve the quality of life in reproductive-aged women [49]. In addition, during menopause and in the postmenopausal period, royal jelly improves genito-urinary syndrome, which is one of the most common complaints of women during these periods [50].

One of the goals of royal jelly supplementation is to improve the quality of life during postmenopausal period. Sharif et al. [51] carried out a study using a cohort of 200 women aged between 45 and 60 years old. Each subject received either capsules of royal jelly (1 g daily) or placebo, during a period of 2 months and the results were compared. The investigators observed that after 8 weeks of royal jelly intake, the menopausal score recorded a significant decrease in comparison with the placebo group. Therefore, royal jelly can alleviate postmenopausal symptoms such as hot flushes, night sweats, and other menopause-related complaints, but the research results on this topic are still inconsistent.

The quality of life during menopause is also disturbed by urinary incontinence or vulvovaginal atrophy, both known as genitourinary syndrome. These complications adversely impact the psychosocial and partner relationships of the postmenopausal women. In order to observe the therapeutic effects of royal jelly for the amelioration of vulvovaginal atrophy, 90 postmenopausal women were randomly distributed to receive vaginal cream of royal jelly, conjugated estrogens or lubricant during a period of 12 weeks [50]. After this period, the vaginal cytology was evaluated among with the quality of life score. The results showed an improvement in the quality of life score of the first group, which received vaginal royal jelly cream. Regarding the vaginal cytology, the Pap smear test showed better results in the group treated with topic estrogens, in comparison with the other two cohorts.

During menopause, sudden estrogen level withdrawal affects the autonomic nervous system, leading to the development of neurodegenerative diseases (e.g., Alzheimer disease), mood disorders (e.g., anxiety, depression) or headache, low back pain or backache. 42 postmenopausal women accusing anxiety, backache and low back pain were recruited and randomly received either 800 mg of dextrin or 800 mg of enzyme-treated royal jelly, for 3 months [52]. After this period, both anxiety score and backache score in royal jelly group were significantly decreased compared to placebo group. Moreover, no side effects were reported during the administration of enzyme-treated royal jelly.

### 3.2. Anticancer Activity

Other beneficial effects of royal jelly consists of the inhibition of tumoral cells growth, tumor-related angiogenesis and activation of immune functions. Cancer is a common pathology in elderly population, and cancer-related fatigue is a complication that further decreases the quality of life of these patients. In these cases, complementary medicine can be a valuable choice in order to ameliorate the symptoms with a minimum of side effects. According to Mofid et al. [53], up to 50% of cancer patients experience cancer-related fatigue, and the administration of both royal jelly and processed honey can alleviate the symptomatology. They administered 5mL of royal jelly and processed honey twice daily to 26 patients, and compared the results with those obtained by 26 patients, which received only 5 mL of honey, under the same scheme, for one month. The conclusion of this clinical trial was that the supplementation of honey intake with royal jelly is more effective for the amelioration of cancer-related fatigue.

In women, the most common cancers are gynecological cancers. In menopause, the most prevalent is breast cancer and royal jelly demonstrated its ability as a complementary therapy in cancer. Bisphenol A (BPA) is one of the most widespread chemicals worldwide, which possesses a structural similarity with estrogens. This characteristic allows BPA to strongly bind to human estrogen-related receptor γ (ERRγ) [54]. In these conditions, BPA is a risk factor for breast cancer in a time-depending and dose-depending manner. Nakaya et al. [55] reported that royal jelly inhibited the growth-promoting effect of BPA in vitro, using MCF-7 cell lines. In the absence of BPA, royal jelly did not affect cellular proliferation.

RJP_30_ is a fraction of royal jelly obtained by precipitation with ammonium sulfate. In vitro, this faction was cytotoxic for human cervical carcinoma cells (HeLa cell line). After only 7 days to the first administration, the cellular density decreased approximately 2,5 fold [56]. Shirzad et al. [57] inoculated tumor cells in 28 male Balb/c mice and after the inoculation, animals orally received various doses of royal jelly (100, 200 or 300 mg/kg). The tumor size was measured every 2 days from the fifth day, and the investigators observed a significant decrease in tumor size. 

### 3.3. Hypocholesterolaemiant Effects

Menopause is a period associated with a high risk for cardiovascular diseases. The main risk factor for the development of cardiovascular pathology is represented by dyslipidemia. Being rich in fatty acids with hypolipidemic properties, royal jelly is used in postmenopausal population in order to restore the balance of the lipidic profile. Despite royal jelly is a potent hypocholesterolaemic and hypolipidaemic agent, its active compounds that lowers the cholesterol levels and the mechanisms of action are still to be discovered. MRJP1 exerts the most potent hypocholesterolaemic effects [43]. MRJP1 interacts with bile acids and enhances the excretion of fecal bile acids. It also induces an increased tendency for fecal excretion of the cholesterol. This major protein of royal jelly also enhances the hepatic cholesterol catabolism. Another hypocholesterolaemiant mechanism of royal jelly consists of the increase of low-density lipoprotein receptor (LDLR) gene expression. This molecule regulates the cholesterol incorporation into the liver. Furthermore, royal jelly decreased gene expression of squalene epoxidase and sterol regulatory element-binding protein in mice, which are the key factors for cholesterol biosynthesis [58].

Lambrinoudaki et al. [59] included 36 postmenopausal individuals in a prospective study and analyzed their circulating cardiovascular risk markers and bone turnover parameters after three months of daily administration of 150 mg royal jelly. No significant changes were recorded in the bone turnover parameters or in the circulating cardiovascular risk markers (protein S, protein C, antithrombin-III and Plasminogen Activator Inhibitor-1). However, significant changes were recorded in terms of the lipid profile. After daily intake of royal jelly, the values of (high-density lipoprotein cholesterol) HDL-C increased, while (low-density lipoprotein cholesterol) LDL-C and total cholesterol (TC) significantly decreased. Therefore, royal jelly supplementation may offer a valuable alternative method of balancing menopause-related dyslipidemic disorders.

In healthy volunteers, the daily intake of royal jelly (6 g/day during 4 weeks) resulted in a significant decrease of LDL-C and TC. No significant differences were observed between cases and controls regarding serum concentrations of HDL-C and TG [60]. A recent meta-analysis [61] reported that royal jelly reduced TC and increased HDL-C serum levels in studies with a long-term follow-up period. The same study mentioned that TG and LDL-C levels were not significantly improved. 

These findings suggest that this natural product may play an important role in cardiovascular health protection by ameliorating the harmful effects of increased cholesterol.

### 3.4. Anti-Hypertensive Effects

Hypertension is one of the most common cardiovascular risk factors in elderly population. It may cause myocardial infarction, heart failure, cerebral stroke, and it is usually associated with metabolic syndrome. Three peptides contained in royal jelly (Ile-Val-Tyr, Val-Tyr, and Ile-Tyr) were reported to inhibit the activity of angiotensin I-converting enzyme and to normalize the systolic blood pressure after only 28 days of treatment in hypertensive rats [62].

Anti-hypertensive and vasodilative effects of royal jelly were studied on hypertensive rats and isolated rabbit thoracic aorta rings [63]. Royal jelly reduced systolic and diastolic blood pressure and increased NO levels in vivo. Furthermore, royal jelly produced vasodilation in isolated rabbit aorta rings. In addition, royal jelly increased the production of nitric oxide (NO)TC and cyclic guanosine monophosphate (cGMP) levels in the aortic rings. In conclusion, anti-hypertensive effects of royal jelly are linked by NO production, and vasodilation is secondary to muscarinic receptor agonists action via NO/cGMP pathway.

### 3.5. Effects on Bone Metabolism

During menopause, bone turnover parameters are usually modified, and osteoporosis is one of the most common menopause-related complications. Bisphosphonates represent the gold standard treatment for postmenopausal osteoporosis. These agents possess a high affinity for bone minerals and bind to hydroxyapatite crystals encountered on the bony surfaces. Their effects consist of the down-regulation of bone resorption mediated by osteoclasts [64]. Royal jelly is a natural alternative used for the prevention of osteoporosis and for the improvement of bone strength and several studies on this topic were realized using animal models.

Royal jelly was administered to ovariectomized rats daily for 3 months, in order to analyze its effects on bone metabolism [65]. After this period, it was reported that femur bone mineral density was not significantly improved. However, the femur stiffness was higher in the study group in comparison to the control group, suggesting that royal jelly cannot prevent the apparition of menopause-related osteoporosis, but it can improve the strength of the bone.

In contradiction with these results, another study [66] sustained that both enzyme-treated royal jelly and royal jelly may prevent osteoporosis. The explanation consists of the enhancing of intestinal calcium absorption. Hidaka et al. administered 17β-estradiol or royal jelly to ovariectomized rats. Their findings showed that royal jelly is almost as effective as hormone replacement therapy in preventing osteoporosis. Furthermore, using a mouse marrow culture model, it was highlighted the inability of royal jelly to inhibit calcium loss induced by the parathyroid hormone and the development of osteoclast-like cells, induced by the same hormone.

### 3.6. Anti-Aging Effects

Royal jelly consumption is associated with an increase in the lifespan of bees and other species. This natural product can delay the apparition of natural aging and some age-related disorders. Furthermore, it has been demonstrated that royal jelly can promote longevity and is useful to improve the quality of life during aging through its components [67]. Proteins and lipids from royal jelly can extend the lifespan of various species, such as honeybees, nematodes, crickets or mice. Xin et al. [68] reported that MRJPs increased the longevity of *Drosophila melanogaster* through the promotion of the pathway mediated by the anti-epidermal growth factor receptor. Moreover, royalactin exerted the same effects on *Caenorhabditis elegans,* by promoting EGF (epidermal growth factor) and EGFR (epidermal growth factor receptor) signaling pathways [69].

Regarding the effects on lifespan extending in mice, studies have shown that long-term intragastric administration of royal jelly prevented aging-related weight loss, improved memory, and delayed aging-related atrophy of the thymus [70]. In addition, the physical performance of the treated animals was better in comparison to the control group: makers of muscle stem cells increased and muscular atrophy was lowered [70]. Furthermore, royal jelly could significantly delay age-related motor functions [67].

Lipid components of royal jelly were reported to inhibit the natural aging process in human cell cultures by two mechanisms: up-regulation of EGF signaling and down-regulation of insulin-like growth factors (IGF) [15]. 10-HDA is the primary lipid compound of royal jelly, and it has been shown to increase the lifespan of *Caenorhabditis elegans* by dietary restriction signaling and TOR components [71]. An interesting question is whether the effects of royal jelly observed on animal model organisms can be extended to humans. Despite there is a small number of studies in this field, the results indicated that anti-aging activity of royal jelly and its compounds on human cell lines, supports previous results reported for model organisms. It has been demonstrated that royal jelly and its special lipidic compound 10-HDA can decrease cellular senescence, and can stimulate the production of procollagen type I and transforming growth factor-β1 (TGF- β1) [72,73]. Moreover, a recent study [74] using human fibroblast cells showed that MRJPs induce higher cellular proliferation, longer telomeres and decreased senescence.

The skin is the largest organ of the body. During menopause and natural aging process, it is affected by the estrogen level withdrawal. Beside, changes in the skin collagen levels leads to a degreased skin elasticity and strength [75]. Royal jelly has been reported as a protective agent against skin aging that acts by enhancing collagen production in ovariectomized rats [76]. It was reported that after the administration of royal jelly 1% to estrogen-deprived females of Sprague-Dawley rats, the level of procollagen type I protein increased in the skin of the animals. Furthermore, the level of recovered collagen was very close to the normal amounts. In conclusion, royal jelly can be considered a valuable and effective dietary supplement against the natural aging process of the skin in postmenopausal period, but future clinical trials on humans are definitely required.

### 3.7. Neuroprotective Effects

Menopause represents a normal milestone that associates a wide range of hormonal changes. In this context, several neurologic illnesses may develop and usually severely impact the psychosocial life of postmenopausal women [77]. Anxiety, depression, or spatial memory loss are some of the most frequent complaints of postmenopausal women. In order to investigate the effects of royal jelly on neurologic disorders, several studies were conducted on animal models.

Due to the aging process, poor mental performance state and neurodegenerative diseases such as Alzheimer’s disease are the most common among the elderly. Royal jelly seems to stimulate the mental functions through its neuroprotective effect. Daily intake of royal jelly was reported to consolidate the memory abilities and learning skills in honeybees and rats [78]. Furthermore, this natural product stimulates neurite outgrowth, induces the regeneration of hippocampal granule cells, and protects the central nervous system against oxidative injuries [33,79].

Royal jelly intake is efficient for the alleviation of menopause-related neurological disorders, but the mechanisms of action remain to be better described. However, the decrease of cholesterol and beta-amyloid levels, the increase of estrogen levels, and the improvement of blood-brain barrier seem to be the most mentioned mechanisms by which royal jelly exerts its neuroprotective role [34]. Zhang et al. [80] showed that purified royal jelly peptides might inhibit β-amyloid 40 and 42 through the regulation of beta-secretase, being useful in ameliorating diseases such as Alzheimer’s disease. Moreover, royal jelly improved the structural image of the brain and behavioral dysfunctions in cholesterol feed rabbits. It actioned through the decrease of body weight, lipid, beta-amyloid and acetyl-cholinesterase levels, associated with the increase of choline acetyltransferase into the brain [80].

Pyrzanowska et al. [81] showed that Greek royal jelly was able to improve the memory performance in aged Wistar rats. They administered to the animals 50 or 100 mg of royal jelly powder/kg twice daily, by gastric gavage, for 8 weeks. After this period, a significant improvement in the memory performance was observed in rats treated with 50 mg powder/kg in comparison to controls. Another study [82] evaluated spatial memory and depression-like behaviors in ovariectomized rats after 82 days of royal jelly ingestion. This natural product improved memory performances and depression-like behaviors. Beside, brain weight and myelin galactolipids slightly increased after royal jelly intake, in comparison to hormone replacement therapy.

### 3.8. Anti-Diabetic Effects

Diabetes mellitus is a very common pathology in the postmenopausal women. In the last years, novel dietary supplements with major therapeutic benefits and a minimum of adverse effects were investigated for the therapeutic management of these patients. In healthy individuals, royal jelly significantly decreased serum glucose levels and increased insulin concentration [83].

Yoshida et al. [84] recently performed a study on obese KK-Ay mice, with type 2 diabetes. 10 mg/kg of royal jelly were administered by oral gavage. Insulin levels, body weight and plasma glucose were analyzed after 4 weeks of daily treatment. The results indicated a partially suppressed body weight associated with decreased glycemia in the study group. However, insulin resistance was not improved. The explanation consists of the suppression of gluconeogenesis by royal jelly, through the decrease of glucose-6-phosphatase mRNA expression. Furthermore, this natural product induced the expression of adiponectin receptor-1, adiponectin, and phosphorylated AMP-activated protein kinase, which generated the suppression of glucose-6-phosphatase, an essential enzyme for gluconeogenesis. Unlike the results obtained by Yoshida et al. [84], who do not mention the decrease in insulin resistance among the effects of royal jelly, Zamami et al. [85] suggested that this dietary supplement could be very effective to prevent insulin resistance. After 8 weeks of daily administration in doses of 100 mg/kg or 300 mg/kg, plasmatic levels of TG and insulin significantly decreased, and systolic blood pressure tended to decrease in ovariectomized rats.

Thirty two male Wistar rats with streptozocin-induced diabetes received 100 mg of royal jelly/kg body weight daily [86]. After six weeks, the urinary levels of urea, creatinine, uric acid, albumin, and histopathological findings of kidneys and liver tissues were significantly improved. Moreover, royal jelly significantly improved the levels of TG, LDL-C, HDL-C, VLDL-C, total cholesterol and ApoA-1 in diabetic subjects, thereby promoting the glycemic balance [87].

Given all the presented studies and findings, we can consider royal jelly a golden dietary supplement, which can be used to alleviate postmenopausal symptoms and to treat a wide range of aging-related pathologies. However, further human clinical trials are necessary in order to better observe beneficial effects, molecular mechanisms of actions and potential side effects of this valuable natural and traditional product. In Table 1 and Table 2, we separately summarized the studies conducted on humans and animals/cells cultures in the last 20 years, in order to investigate the efficiency of royal jelly as a remedy for postmenopausal problems and complications.

Figure 3 is a schematic representation of the royal jelly’s beneficial effects in relieving menopause symptoms and aging-related diseases.

## 4. Side Effects of Royal Jelly

Royal jelly is a natural product widely consumed worldwide. However, several side effects of its consumption have been described. The major allergens of this product are MRPJ 1 and MRPJ2 [100], and these are common with honeybee venom allergens. MRJPs can cause several allergic reactions such as asthma, dermatitis, skin rashes, eczemas, bronchospasm, anaphylaxis, hemorrhagic colitis [101], or even anaphylactic shock and death, in some situations [102]. A positive diagnosis of royal jelly anaphylaxis is established based on a positive prick test and systemic clinical symptoms. Oral allergy syndrome secondary to royal jelly intake has also been mentioned in the literature [103].

No reports regarding the possible interactions between royal jelly supplements and other drugs concomitantly taken were made. However, Lee et al. [104] presented the case of an 87-years-old man with long-term warfarin therapy, supplemented with royal jelly capsules. He referred to the hospital for hematuria, and the most probable explanation for his symptoms was a possible interaction between warfarin and royal jelly. For this reason, clinicians should repeatedly inform their patients regarding potential drug interactions with dietary supplements.

## 5. Conclusions and Future Perspectives

Royal jelly is widely used for various medical and commercial purposes. Its biologically active compounds lend this natural product various pharmaceutical applications, with very few side effects. Accumulating evidence from the most recent studies indicated that royal jelly possesses a significant role in modulating the mechanisms of natural aging. This product has demonstrated its efficiency in the alleviation of postmenopausal complaints. Furthermore, it could be successfully used by postmenopausal women as a dietary supplement for the treatment of aging-related pathologies, such as neurodegenerative diseases, diabetes, obesity, cancer, osteoporosis or cardiovascular pathology. Unfortunately, there is only a small amount of clinical trials on humans regarding the effects of royal jelly in the postmenopausal period, the main majority being conducted on animal models. We consider that this paper paves the way for further investigations in the field. In our opinion it is very important to investigate whether and through which mechanisms royal jelly could improve sleep disorders, hot flushes, postmenopausal headache, anxiety or depression. Furthermore, the doses and the period of administration represent a topic of maximum interest because it can significantly influence the final results. Therefore, further clinical trials are imperative in order to elucidate the real value of this natural product as food supplement during menopause.

## Figures and Tables

**Figure 1 molecules-25-03291-f001:**
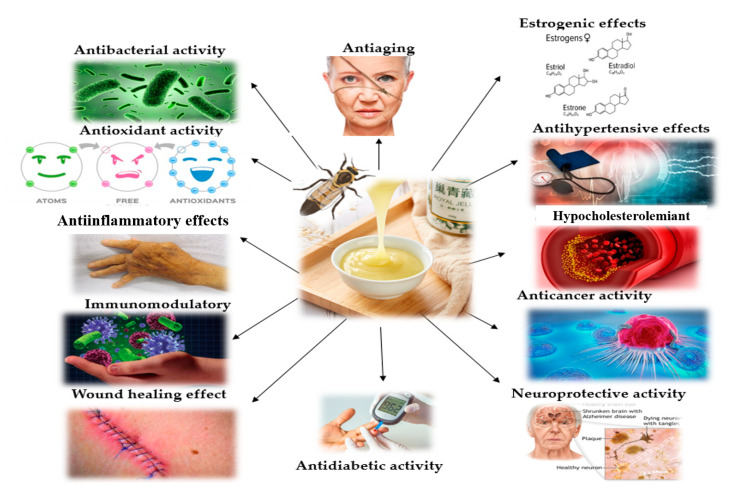
Biological properties of royal jelly [9].

**Figure 2 molecules-25-03291-f002:**
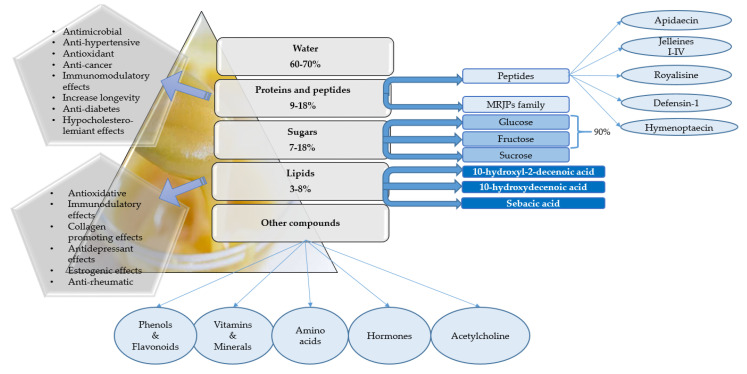
Schematical representation of the biological composition of royal jelly and the main functional activities of the compounds [37,38,39,40,41,42,43].

**Figure 3 molecules-25-03291-f003:**
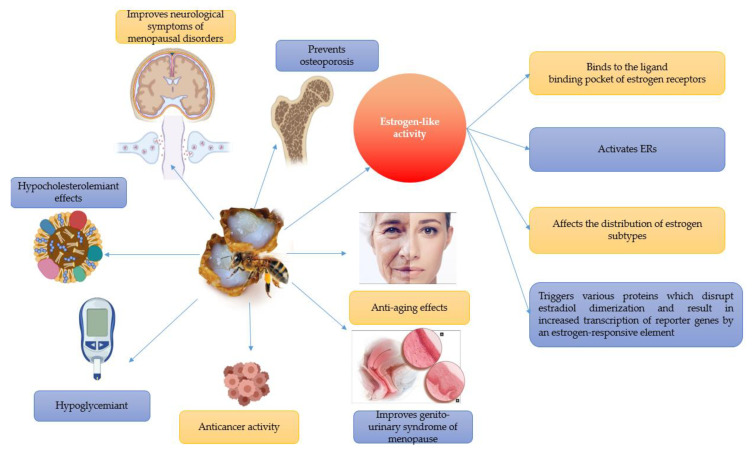
Royal jelly’s beneficial effects in relieving menopause symptoms and aging-related diseases. This figure also illustrates the essential pathways through which royal jelly exerts its estrogen-like action, one of the most important characteristics of this natural product.

**Table 1 molecules-25-03291-t001:** Human studies regarding beneficial effects of royal jelly in menopause.

*Author, Reference*	*Beneficial Effect*	*Study Design*	*Number of Subjects*	*Dose and* *Administration*	*Results*
*Kaczor et al., 1962* *[88]*	Anti-atherosclerosis	Randomized, placebo-controlled study	27 patients and 12 controls	Seven patients: 100 mL subcutaneously every day for 2 weeks, and 4 weeks on alternate days 20 patients: same scheme for 3 weeks Controls: placebo—3 weeks	In supplemented patients - Cholesterol levels decreased - Phospholipid levels increased - Total lipid levels insignificantly decreased
*Guo et al., 2007* *[60]*	Improves lipoprotein metabolism	Randomized, placebo-controlled study	15 patients	Seven cases received 6 g of royal jell daily, for 4 weeks Eight cases were the control group	- TC and LDL-colesterol significantly decreased in comparison to the control group - No significant differences were noted in HDL-colesterol and TG serum concentrations
*Mobasseri et al., 2014 [89]*	Hypolipemiant	Randomized, placebo-controlled study	50 women (25 cases and 25 controls) with type 2 diabetes	1000 mg/day of royal jelly or placebo—8 weeks	In supplemented patients - Serum triglyceride levels and total cholesterol significantly decreased - HDL-c insignificantly increased in both groups - hs-CRP (high-sensitivity C-reactive protein) significantly decreased, while in the control group it remained elevated
*Mobasseri et al., 2015 [90]*	Hypoglycemiant	Randomized, placebo-controlled study	50 patients (20 cases and 20 controls) with type 2 diabetes	10 g of fresh royal jelly or placebo after 12 h fasting	- The mean serum glucose concentration significantly decreased in both groups - The mean insulin concentrations were after one hour, and insignificantly decreased after 2 h of royal jelly - no significant difference was observed in the glycemic control parameters between both studied groups
*Shidfar et al., 2015 [91]*	*Anti-diabetic*	*Randomized, placebo-controlled study*	*46 type 2 diabetic patients*	*1000 mg royal jelly or placebo—3 times/day for 8 weeks*	- *In supplemented patients, HOMA-IR significantly decreased and total antioxidant capacity increased* - *Serum insulin concentrations were not significantly different between the groups*
*Mofid et al., 2016 [53]*	Anti-fatigue	Randomized, double-blind placebo-controlled study	52 cancer patients	26 patients received 5 mL of processed honey and royal jelly twice/day—4 weeks 26 controls received 5 mL of pure honey twice/day—4 weeks	- After 2 weeks and 4 weeks of treatment, the scores on visual analogue fatigue scale and fatigue severity scale were significantly improved in the first group, supplemented with royal jelly and processed honey
*Chiu et al., 2016 [92]*	Hypocholesterolemiant Decreases the risk of cardiovascular disease	Randomized, placebo-controlled study	40 patients with mild hypercholesterolemia	Nine capsules of royal jelly or placebo, daily—3 months (350 mg of royal jelly or placebo /capsule)	- In supplemented group, LDL-c and TC levels significantly improved after 3 months - TG and HDL-C were not significantly altered in the royal jelly group - The concentration of dehydroepiandrosterone sulphate (DHEA-S) was ameliorated after three months in the supplemented group
*Lambrinoudaki et al., 2016 [59]*	Improves the lipid profile	Prospective study	36 postmenopausal women	150 mg/day, 3 months	- Royal jelly intake significantly increased HDL-C levels and decreased TC and LDL-C
*Seyyedi et al., 2016 [50]*	Improves sexual and urinary function	Randomized, placebo-controlled study	90 postmenopausal women	Group 1—vaginal cream of royal jelly 15%, 3 months Group 2—lubricant, 3 months Group 3—conjugated estrogens, 3 months	- Royal jelly was superior to conjugated estrogens and lubricants for the improvement of QoL and genitourinary syndrome in postmenopausal women - Improvement of vaginal atrophy in conjugated estrogens group was considerably more effective than royal jelly or lubricant
*Asama et al., 2018 [52]*	Improves backache, low back pain and anxiety	Randomized, placebo-controlled study	42 postmenopausal women	800 mg of enzyme-treated royal jelly or 800 mg of dextrin—3 months	- After 12 weeks of treatment with enzyme-treated royal jelly significant differences were observed in anxiety score, low back pain and backache scores, in comparison to placebo group.
*Sharif et al., 2019 [51]*	*Improves menopause-related symptoms*	*Randomized, double-blind placebo-controlled study*	*200 postmenopausal women*	*1000 mg royal jelly or placebo/day—8 weeks*	- *The mean baseline menopausal score was similar in both groups, but in royal jelly group it was significantly reduced after 8 weeks of intake*
*Petelin et al., 2019 [93]*	Improves the lipid profile, satiety and antioxidant capacity	Randomized, double-blind placebo-controlled study	60 obese patients (30 cases and 30 controls)	Two capsules of lyophilized royal jelly (333 mg/capsule) or placebo—8 weeks	- Royal jelly significantly decreased TC and inflammatory marker C-reactive protein - Royal jelly increased adiponectin, leptin and total antioxidant capacity in serum

**Table 2 molecules-25-03291-t002:** Studies regarding beneficial effects of royal jelly in menopause using animal models and cellular lines.

*Author, Reference*	*Beneficial Effect*	*Study Design*	*Subjects*	*Dose and* *Administration*	*Results*
*Hidaka et al., 2006 [66]*	Prevents osteoporosis	Prospective study	48 female Sprague-Dawley rats (6 controls and 42 ovariectomized rats) with low tibial bone mineral density	The rats were divided into 8 groups and royal jelly was mixed with MF powdered pellets, in order to be orally administered (0.5 g of royal jelly mixed with 100 g MF pellets or 2 g of royal jelly mixed with 100 g MF pellets).	- The administration of 2% royal jelly and 0.5–2% royal jelly to ovariectomized rats recovered more than 85% of the tibial bone mineral density. - Royal jelly proved its efficiency in the prevention of osteoporosis
*Zamami et al., 2008 [85]*	Improves insulin resistance	Randomized, Placebo-controlled study	Male Winstar rats	100 or 300 mg/kg, po—8 weeks or placebo	- Insulin and TG levels significantly increased in royal jelly group - Systolic blood pressure was decreased in royal jelly group - Blood glucose concentration and TC were not affected by royal jelly intake
*Takaki-Doi et al., 2009* [94]	*Anti-hypertensive*	*Prospective study*	*Hypertensive rats*	*Seven peptide fractions of royal jelly protein hydrolysate were administered separately in doses of 10, 30, 100 mg/kg i.v. or 1000 mg/kg po*	- *Both royal jelly protein hydrolysate and different peptide fractions of royal jelly protein hydrolysate showed potent hypotensive effects in comparison to placebo group*
*Park et al., 2012* [76]	Anti-aging effects on the skin	Prospective study	Ovariectomized virgin female of Sprague-Dawley rats	The rats were fed with a dietary supplement containing 1% royal jelly extract	- The collagen content and epidermal thickness of skin tissue were measured - The investigators observed increased level of procollagen type I protein in the dorsal skin of the rats
*Zamani et al., 2012* [78]	Neuroprotective role	Placebo-controlled study	Rats bilaterally intracerebroventricular infused with streptozocin	- The rats included in the study group were supplemented with royal jelly, while the control group rats were feed with regular food - The spatial learning and memory were tested using Morris water maze test	- Royal jelly showed improved memory in the study group, supporting the hypothesis that it can exert helpful effects in Alzheimer’s disease
*Shirzad et al., 2013* [57]	Decreases the sizes of malignant tumors	Prospective study	28 male Balb/c mice subcutaneously injected with tumor cells	Group 1 received 100 mg/kg of royal jelly Group 2 received 200 mg/kg of royal jelly Group 3 received 300 mg/kg of royal jelly Group 4 received a vehicle	- The tumoral size was significant smaller in case groups, in comparison to controls.
*Pyrzanowska et al., 2014 [81]*	Improves spatial memory	Randomized, placebo-controlled study	18-month old male Winstar rats	50 and 100 mg of royal jelly powder/kg/day by gastric gavage—8 weeks	- Significant improvements of memory in rats treated with 50 mg/kg/day of royal jelly were observed compared with controls
*Kaku et al., 2014 [95]*	Improves bone quality	Prospective study	Ovariectomized rats	Royal jelly was administered to the rats for 12 weeks, orally	- Royal jelly intake did not influence bone volume but improved bone quality by modulating type I collagen.
*Minami et al., 2016 [82]*	*Improves neurological symptoms of menopausal disorders*	*Randomized, placebo-controlled study*	*Ovariectomized rats*	*Royal jelly was administered to the rats for 82 days*	- *Royal jelly impaired memory and depressive mood in ovariectomized rats* - *Brain weight was significantly increased by royal jelly intake, in comparison to E2 administration* - *Proteins and galactolipids were more increased in the brain of the rats from royal jelly group than in E2 group*
*Yoshida et al., 2016 [84]*	Hypoglicemiant	Randomized, placebo-controlled study	Obese/diabetic KK-Ay mice	10 mg/kg of royal jelly daily, by oral gavage—4 weeks	- Royal jelly administration improved hyperglycemia and partially suppressed body weight gain, but insulin resistance was not influenced
*Chen et al., 2017* *[96]*	Improves spatial memory	Randomized, placebo-controlled study	Male aged rats	The rats were supplemented for 14 weeks with royal jelly. Controls received distilled water	- The male rats fed with MRJPs improved their spatial memory with more than 48,5% in comparison to control group
*Shimizu et al., 2018* *[65]*	Improves bone quality	Randomized, placebo-controlled study	12 weeks old ovariectomized rats	Royal jelly was administered daily, for 3 months	- Royal jelly significantly improved femur stiffness in ovariectomized rats
*Jiang et al., 2018* *[74]*	Anti-senescence activity	Prospective study	Human embryonic lung fibroblast cells (HFL-I)	HFL-I cells were cultured in media containing various concentrations of MRJPs	- MRJPs exerted anti-aging activity for the HFL-I cell line
*Sefirin et al., 2019* *[97]*	Anxiolytic effectsPrevents hot flushes	Randomized, placebo-controlled study	Ovariectomized Winstar rats	100, 200 and 300 mg/kg of royal jelly	- Royal jelly significantly decreased the frequency of hot flushes in ovariectomized mice, and exerted anxiolytic effects - In the Open Field test, royal jelly increased the centre square time and the number of rearing - Royal jelly induced a significant increase of the number of rearing and head dipping in the Elevated Plus-Maze test
*Liu et al., 2019* *[98]*	*Protects reproductive function*	*Randomized, placebo-controlled study*	*50 female mice*	*Group 1: 125, 250, 500 mg/kg MRJPs daily* *Group 2: 125 mg/kg casein daily* *Group 3: saline solution dailyIntragastric administration for seven weeks*	- *Royal jelly supplement (medium and high dosages) increased uterus and ovarian index.* - *Serum levels of E2 and progesterone were significantly increased in the MRJPs groups* - *The levels of FSH and LH were significantly decreased in royal jelly groups in comparison with control group.* - *Follicular development was also improved in the groups treated with MRJPs*
*Pan et al., 2019* *[63]*	Antihypertensive effects	Randomized, placebo-controlled study	Hypertensive rats	Group 1: 1g/kg of royal jelly, by oral administration, for 4 weeks Group 2: control group	- Heart rate, systolic blood pressure and diastolic blood pressure were significantly decreased in the study group after 4 weeks of treatment, in comparison to the control group.
*You et al., 2020* *[99]*	Attenuates nonalcoholic fatty liver disease	Prospective study	Ovariectomized rats	150, 300 or 450 mg/kg/day for 2 months	- RJ improved the anxiety level, normalized serum lipid profile, and attenuated the non-alcoholic hepatic steatosis and liver injury.

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
