# Peer review of "Royal Jelly—A Traditional and Natural Remedy for Postmenopausal Symptoms and Aging-Related Pathologies"

_molecules, 2020, doi:10.3390/molecules25143291_

Round 1
Reviewer 1 Report
The manuscript entitled “Royal Jelly – A Traditional and Natural Remedy for Postmenopausal Symptoms and Aging-Related Pathologies” is an interesting review manuscript.
However, I believe that the Authors have not taken into consideration a very recent review published in the journal “International Journal of Molecular Sciences, of which, however, they follow the structure: Ahmad, S., Campos, M. G., Fratini, F., Altaye, S. Z., & Li, J. (2020). New insights into the biological and pharmaceutical properties of royal jelly. International Journal of Molecular Sciences, 21(2), 382.
Authors may indeed take inspiration for some sections of their review from this recent publication, but I would advise them to change much of the structure of their review.
In detail I would propose to focus attention only and exclusively on postmenopausal disorders and aging-related pathologies by going deeper into the two chapters and dividing them into specific paragraphs; moreover, I suggest to completely remove or otherwise reduce to essentials all the first part of the review in which they deal with the different biological properties of royal jelly that are excessive being another the main focus of the work. I would rather insert an accurate meta-analysis of the scientific literature available on these two aspects accompanied by one or two tables in which all references are reported so that the reader has a useful and easy and immediate consultation tool.
Finally, I propose to the Authors to make all these structural changes in order to better characterize their review and make it original compared to the other, avoiding a plethora of publications too similar on such an important focus.
Author Response
Dear Sir or Madam,
We thank you for your cooperation and we appreciate you taking the time to analyze our work. Following the comments and suggestions, we made some major revisions to our paper, in order to better characterize our review.
- We reduced to essentials the first part of our paper, regarding the chemical structure of royal jelly.
- We completely removed the chapter regarding potential beneficial activities of royal jelly, and we focused on the effects of this product on relieving menopause symptoms and aging-related diseases.
- All the reviewed studies were summarized in chronological order of their publication in Table 1 (human studies) and Table 2 (animal models and studies conducted on cells lines).
- Figure 3 was realized in order to summarize the most important effects of royal jelly on relieving menopause symptoms and aging-related diseases, as they were presented and described in text.
- We introduced in the last chapter “Conclusions” some future perspectives and some personal opinion regarding the topic of our paper.
After analyzing the revised manuscript we feel confident that, without being all encompassing our study is significant and pertinent to medical researchers.
We thank you for your time and cooperation in this matter and we really appreciate your efforts.

Reviewer 2 Report
Dear Authors,
The present study ID: molecules-863922 entitled "Royal Jelly – A Traditional and Natural Remedy for Postmenopausal Symptoms and Aging-Related Pathologies" written by authors Andreea Balan , Marius Alexandru Moga , Lorena Dima , Sebastian Toma , Andrea Elena Neculau , Costin Vlad Anastasiu is devoted to royal jelly in the view of biological a medicinal function overall.
This review showed significant information about royal jelly - composition, beneficial action, and text about therapeutic properties for postmenopausal symptoms and aging-related diseases. The text is properly organized and includes very interesting and important information for readership.
I have some suggestion in order to improve manuscript, which are the following:
- It is better to use the correct hyphen in several places in the text - eg. L.169 (80-85%), L. 172, 191, 192, 197, etc. (along the whole paper).
- It is advisable to use a subscript when labeling vitamins. - eg. L- 198-199, etc.
- Why do the authors box the labels in Figure 2 with different font sizes? Please, correct.
- Correct taxonomic of Salmonella - see L. 254.
- Different citations are used in the text (inconsistent) - see L. 282 (...and collab...), L. 301, 337, etc. along the paper.
- The chapter 4 should be indented from the previous text.
- Table 1, etc. - please, correct, according to authors manual of Molecules.
- The citation style is not completely uniform according to the requirements of Molecules.
Author Response
Dear Sir or Madam,
We thank you for your cooperation and we appreciate you taking the time to analyze our work. Following the comments and suggestions, we made some major revisions to our paper, in order to better characterize our review.
- We replaced the hyphen all along the paper
- We used subscripts for vitamins
- We adjusted the font sizes in Figure 2
- We corrected the citations and replaced “and collab”/”and coworkers” with “et al.”
- The citation errors were solved
- We reduced to essentials the first part of our paper, regarding the chemical structure of royal jelly.
- We completely removed the chapter regarding potential beneficial activities of royal jelly, and we focused on the effects of this product on relieving menopause symptoms and aging-related diseases.
- All the reviewed studies were summarized in chronological order of their publication in Table 1 (human studies) and Table 2 (animal models and studies conducted on cells lines).
- Figure 3 was realized in order to summarize the most important effects of royal jelly on relieving menopause symptoms and aging-related diseases, as they were presented and described in text.
- We introduced in the last chapter “Conclusions” some future perspectives and some personal opinion regarding the topic of our paper.
Please be more specific regarding the modifications you suggest for table 1. We realized it according to authors instructions of Molecules Journal.
After analyzing the revised manuscript we feel confident that, without being all encompassing our study is significant and pertinent to medical researchers.
We thank you for your time and cooperation in this matter and we really appreciate your efforts.

Reviewer 3 Report
This study aimed to highlight the effectiveness of royal jelly supplementation in relieving menopause symptoms and aging-related diseases. Although the activities of royal jelly have been found from numerous aspect, the activity of relieving menopause symptoms and aging-related diseases is definitely limited. I suppose, however, the focus level on the activity of relieving menopause symptoms and aging-related diseases should be improved, in order to understand their relationship. Therefore, there are some points as following:
(1) The potential beneficial actions of royal jelly, such as antibacterial effects, anti-aging, neuroprotective effects…, have been listed and stated briefly as one part. However, this part is not the topic in this review. Suggest that the widely potential functions of royal jelly may be concentrated to highlight the activity of royal jelly in relieving menopause symptoms and aging-related diseases.
(2) It was always mentioned: Author showed that ….. For example, L282, “Ji and collab. showed that long-term…”; L286, “Okumura et al. also analyzed the effects of enzyme…”; L298, “Two studies realized by Zheng and Park pointed out that royal jelly…”. Suggest that some points need to be summarized and put forward your own views throughout of the manuscript.
(3) In Table 1, only human studies regarding beneficial effects of royal jelly in menopause involved. Please also summarized the cell and animal trials mentioned in the manuscript in Table 1.
(4) It is appreciated to make a summary of mechanism in relieving menopause symptoms and aging-related diseases with a figure. The figure should cover the pathways and approaches described in text.
(5) It is appreciated to provide the future perspectives of the effectiveness of royal jelly on relieving menopause symptoms and aging-related diseases in Conclusion part.
Author Response
Dear Sir or Madam,
We thank you for your cooperation and we appreciate you taking the time to analyze our work. Following the comments and suggestions, we made some major revisions to our paper, in order to better characterize our review.
- We reduced to essentials the first part of our paper, regarding the chemical structure of royal jelly.
- We completely removed the chapter regarding potential beneficial activities of royal jelly, and we focused on the effects of this product on relieving menopause symptoms and aging-related diseases.
- All the reviewed studies were summarized in chronological order of their publication in Table 1 (human studies) and Table 2 (animal models and studies conducted on cells lines).
- Figure 3 was realized in order to summarize the most important effects of royal jelly on relieving menopause symptoms and aging-related diseases, as they were presented and described in text.
- We introduced in the last chapter “Conclusions” some future perspectives and some personal opinion regarding the topic of our paper.
- English language was also edited.
After analyzing the revised manuscript we feel confident that, without being all encompassing our study is significant and pertinent to medical researchers.
We thank you for your time and cooperation in this matter and we really appreciate your efforts.

Round 2
Reviewer 1 Report
The manuscript has been deeply revised and corrected by the Authors taking into due consideration every suggestion of the referees.
However, some important scientific publications that could have been used both for the "Intorduction" section and for the discussion of some parts have not been included as previously recommended.
I therefore invite the Authors to insert them and for simplicity and completeness I report them here below:
Fratini, F., Cilia, G., Mancini, S., & Felicioli, A. (2016). Royal Jelly: An ancient remedy with remarkable antibacterial properties. Microbiological Research, 192, 130-141.
Ahmad, S., Campos, M. G., Fratini, F., Altaye, S. Z., & Li, J. (2020). New insights into the biological and pharmaceutical properties of royal jelly. International Journal of Molecular Sciences, 21(2), 382.
Author Response
Dear Sir or Madam,
According to your suggestions, we refereed the following studies:
- Fratini, F., Cilia, G., Mancini, S., & Felicioli, A. (2016). Royal Jelly: An ancient remedy with remarkable antibacterial properties. Microbiological Research, 192, 130-141.
- Ahmad, S., Campos, M. G., Fratini, F., Altaye, S. Z., & Li, J. (2020). New insights into the biological and pharmaceutical properties of royal jelly. International Journal of Molecular Sciences, 21(2), 382.
Thank you for you cooperation and for your comments!
Reviewer 3 Report
The author has been revised most of the suggestion. However, some points still need to be noted.
(1) L104-L109, L110-L115, L120-126, L127-129, L174-176, L178-179: This information should be remained.
(2) In the section of “Hypocholesterolaemiant Effects”, please state the relationship between cardiovascular diseases and Postmenopausal Symptoms/Aging-Related Diseases at the beginning of this part.
(3) In the section of “Anti-Diabetic Effects”, please state the relationship between diabetes and Postmenopausal Symptoms/Aging-Related Diseases at the beginning of this part.
(4) It is appreciated that Figure 3 was drew to summarize the beneficial effects in relieving menopause symptoms and aging-related diseases. If it is possible, please add the signaling pathway related to the discussed diseases in detail.
Author Response
Dear Sir or Madam,
According to your suggestions, we made the following modifications:
- L104-L109, L110-L115, L120-126, L127-129, L174-176, L178-179 were introduced again
- The relationship between cardiovascular diseases/diabetes and Postmenopausal Symptoms/Aging-Related Diseases was highlighted at the beginning of each of the following sections: “Hypocholesterolaemiant Effects” and “Anti-Diabetic Effects”
- The main signaling pathways related to the discussed affections were illustrated in Figure 3.
We thank you for your cooperation and for you valuable comments!